# Oleuropein Aglycone Peracetylated (3,4-DHPEA-EA(P)) Attenuates H_2_O_2_-Mediated Cytotoxicity in C2C12 Myocytes via Inactivation of p-JNK/p-c-Jun Signaling Pathway

**DOI:** 10.3390/molecules25225472

**Published:** 2020-11-23

**Authors:** Monica Nardi, Sara Baldelli, Maria Rosa Ciriolo, Paola Costanzo, Antonio Procopio, Carmela Colica

**Affiliations:** 1Dipartimento di Scienze Della Salute, Università Magna Graecia, Viale Europa, 88100 Germaneto, Italy; pcostanzo@unicz.it (P.C.); procopio@unicz.it (A.P.); 2Department of Human Sciences and Promotion of the Quality of Life, IRCCS San Raffaele Pisana, San Raffaele Roma Open University, 00163 Rome, Italy; 3Department of Biology, University of Rome “Tor Vergata”, 00133 Rome, Italy; ciriolo@bio.uniroma2.it; 4IRCCS San Raffaele Pisana, 00163 Rome, Italy; 5CNR, IBFM UOS, Università Magna Graecia, Viale Europa, 88100 Germaneto, Italy; carmela.colica@cnr.it

**Keywords:** 3,4-DHPEA-EA derivatives, C2C12 myocytes, olive oil, antioxidant, skeletal muscle

## Abstract

Oleuropein, a glycosylated secoiridoid present in olive leaves, is known to be an important antioxidant phenolic compound. We studied the antioxidant effect of low doses of oleuropein aglycone (3,4-DHPEA-EA) and oleuropein aglycone peracetylated (3,4-DHPEA-EA(P)) in murine C2C12 myocytes treated with hydrogen peroxide (H_2_O_2_). Both compounds were used at a concentration of 10 μM and were able to inhibit cell death induced by the H_2_O_2_ treatment, with 3,4-DHPEA-EA(P) being more. Under our experimental conditions, H_2_O_2_ efficiently induced the phosphorylated-active form of JNK and of its downstream target c-Jun. We demonstrated, by Western blot analysis, that 3,4-DHPEA-EA(P) was efficient in inhibiting the phospho-active form of JNK. This data suggests that the growth arrest and cell death of C2C12 proceeds via the JNK/c-Jun pathway. Moreover, we demonstrated that 3,4-DHPEA-EA(P) affects the myogenesis of C2C12 cells; because MyoD mRNA levels and the differentiation process are restored with 3,4-DHPEA-EA(P) after treatment. Overall, the results indicate that 3,4-DHPEA-EA(P) prevents ROS-mediated degenerative process by functioning as an efficient antioxidant.

## 1. Introduction

Oleuropein, the main phenolic compound of olive leaves, is an important bioactive compound with various biological properties, including anticancer [1], antidiabetic and antiatherosclerotic [2]. Moreover, several in vitro and in vivo studies show that oleuropein is a potent scavenger of superoxide anion radicals and of other reactive oxygen species (ROS) [3].

Among the oleuropein derivatives, isomeric forms of its aglycone (3,4-DHPEA-EA), a hydrolysis product of oleuropein, has been shown to have important biological functions among which: reducing lung inflammation in a mouse-model of carrageenan-induced pleurisy [4], reducing the plasma levels of pro-inflammatory cytokines, ameliorating the development of collagen-induced arthritis [5] and improving the function which protects from Aβ deposition [6]. It has also been shown that semisynthetic peracetylated oleuropein and its peracetylated derivatives improve the capacity to permeate cell membrane, with enhanced biological activity [2,7]. Furthermore, it has been reported that the peracetylation of oleuropein not only improves its affinity to fatty matrix food such as extra-virgin olive oil (EVOO), but also the stability of the peracetylation of oleuropein, also over time after the enrichment [8].

In recent years, the importance of the antioxidant capacity of 3,4-DHPEA-EA and its derivatives [9,10] in relation to skeletal muscle has emerged [11,12,13]. Skeletal muscle is a tissue in which ROS are of particular importance: at low concentrations ROS act as signaling molecules in signal transduction pathways; instead, at high concentrations they can induce oxidative stress and muscular atrophy [14]. In this context, many natural molecules have been tested for their antioxidant properties and the faculty to reduce ROS production in skeletal muscle [15]. Indeed, pathophysiological conditions such as sarcopenia, muscular atrophy and strenuous exercise are characterized by an increase in radicals [16,17,18], which can be buffered or prevented through the use of natural molecules such as resveratrol [19] and plant extracts [20].

Among the natural molecules, 3,4-DHPEA-EA and its derivatives seem to play an important role in the regulation of skeletal muscle homeostasis [11,12]. Specifically, it has been demonstrated that oleuropein induces an activation of AMP-activated protein kinase (AMPK), with a concomitant increase in GLUT4 translocation at the cell membrane and glucose uptake [12,21]. Oleuropein also prevents palmitic acid-induced myocellular insulin resistance, suggesting the possibility for this molecule to be active for type 2 diabetes by decreasing insulin levels at muscular level [11]. Moreover, it has been shown that derivatives of oleuropein reduce high fat diet-induced lipid deposits in liver and skeletal muscle, enhance enzymatic antioxidant activity, modulate the synthesis of mitochondrial complex subunits and eventually inhibit apoptosis activation [22,23]. Despite such evidence, the molecular mechanism(s) that characterize and mediate the activity of natural molecule oleuropein derivatives in skeletal muscle in conditions of homeostasis or after the alteration of the redox state, as well as during muscular atrophy are currently poorly addressed. Studies presented in the literature have shown that H_2_O_2_, the most stable ROS, stimulates different biological responses in the skeletal muscle, ranging from physiological responses to detrimental effects [24,25]. In particular, high doses of H_2_O_2_ reduced myocyte viability and impaired energetic metabolism [24,25,26]. Nevertheless, the molecular mechanism(s) that underlie(s) the H_2_O_2_-induced alteration in the intracellular redox state, and the cell viability of myocytes, is/are still undefined.

Many molecules have been used to maintain, prevent or treat oxidative stress at skeletal muscle level, but in recent years a particular interest has turned towards the beneficial effects of oleuropein, especially at muscular level. As previously mentioned, oleuropein has many biochemical functions, including antimicrobial, anticancer and, most of all, antioxidant [27,28]. Despite this evidence, the exact molecular mechanism that mediates the antioxidant effects of oleuropein in the skeletal muscle has been little studied so far.

Thus, in this work, we analyzed the antioxidant effect of low dose of aglycone peracetylated (3,4-DHPEA-EA(P), obtained through extraction of oleuropein, followed by synthesis of its aglycone and subsequent peracetylation (Scheme 1), in murine C2C12 myocytes, treated with hydrogen peroxide (H_2_O_2_).

We demonstrated that the cytotoxic effects of H_2_O_2_ on the myocytes’ viability and myogenesis proceeds via the activation of a p-JNK-p-c-Jun pathway, which are abolished by 3,4-DHPEA-EA(P) treatment. Moreover, we demonstrate that oxidative stress induces a decrease in myogenic transcription factor MyoD levels indicating a delay in the differentiation process and an induction of muscular atrophy. Thus, our data show the importance of using natural molecules such as 3,4-DHPEA-EA(P) in preventing degenerative processes by protecting against hydrogen peroxide challenge.

## 2. Results

### 2.1. H_2_O_2_ Reduces C2C12 Myocytes Viability and Myogenesis

In the present work, we studied the effects of 3,4-DHPEA-EA(P), a oleuropein peracetylated derivative with various pharmacological functions [7], on C2C12 myocytes challenged with H_2_O_2_. For this purpose, C2C12 myocytes were differentiated for 2 days and treated with different concentrations of H_2_O_2_ for 24 h.

Figure 1A shows that H_2_O_2_ significantly affects cell viability as determined by MTS assay, in a concentration-dependent manner from 20 μM to 100 μM. Thus, a concentration of 100 μM H_2_O_2_ was selected for all subsequent experiments. To characterize this effect more precisely, we performed a direct cell count following the Trypan blue staining, 24 h after the treatment with 100 μM of H_2_O_2_. Figure 1B shows that the number of viable cells was profoundly affected, with a concomitant increase in dead cells (Figure 1C). These data were also validated by BrdU immunochemistry, which showed a significant decrease of BrdU incorporation in H_2_O_2_-treated myocytes after 24 h, also implying a cell cycle arrest (Figure 1D). The H_2_O_2_-mediated cytotoxic effect was also established in terms of myogenesis. Indeed, the H_2_O_2_ treatment induced an alteration in the differentiation program, with an inhibition of MyoD expression (Figure 1E), a marker of the early stages of skeletal muscle differentiation [29]. To assess that the cytotoxic effect on the myocyte viability caused by H_2_O_2_, we treated C2C12 cells with catalase, which scavengers H_2_O_2_. Figure 1B demonstrates that the catalase treatment effectively counteracts the H_2_O_2_-mediated decrement in the number of myocytes, which was completely restored as in untreated conditions.

### 2.2. 3,4-DHPEA-EA(P) Prevents the Cytotoxic Effect of H_2_O_2_ in C2C12 Myocytes

Next, we tested whether oleuropein derivatives could function as antioxidant agent and mitigate H_2_O_2_-mediated cytotoxicity in C2C12 myocytes. Firstly, we carried out a concentration response curve with H_2_O_2_ and oleuropein derivatives at different concentrations (5, 10, 50 and 100 μM) for 24 h, in order to choose the optimal dose capable of reversing the decrease in C2C12 cell viability. In particular, C2C12 myocytes at day 2 of differentiation were pretreated with 5, 10, 50 and 100 μM 3,4-DHPEA-EA and 3,4-DHPEA-EA(P) for 1 h, and subsequently with 100 μM H_2_O_2_ for 24 h.

Figure 2A shows that the two natural compounds are able to recover cell viability after the treatment with H_2_O_2_, even though with different degrees of efficiency. In fact, we have chosen a concentration of 10 μM for both compounds, as it seemed to be the best to achieving a recovery of cell viability. Figure 2B,C show that pre-treatment with 3,4-DHPEA-EA(P) was significantly more capable at offsetting the growth arrest and cell death rise caused by the H_2_O_2_ treatment, as assessed through the cell count by Trypan blue staining and BrdU incorporation (Figure 2D). The same result was obtained with an MTS analysis (Figure 2E), confirming that 3,4-DHPEA-EA(P) exerted a protective effect against the H_2_O_2_-induced cytotoxicity. Thus, for subsequent experiments we chose the compound which better re-established cell viability, i.e., 3,4-DHPEA-EA(P), in order to highlight the molecular mechanism(s) that mediate the protective effect.

### 2.3. 3,4-DHPEA-EA(P) Counteracts H_2_O_2_-Mediate Oxidative Stress Damage in C2C12 Myocytes

We then asked ourselves whether 3,4-DHPEA-EA(P) was able to counteract the deleterious effects of H_2_O_2_ on the C2C12 cell number by buffering intracellular ROS. The intracellular ROS levels were evaluated by means of cytofluorimetric analysis, using the ROS-sensitive probe: DCF-DA and DHE. Figure 3A,B shows that 10 μM 3,4-DHPEA-EA(P) was able to lower the intracellular ROS content. In addition, we measured the level of Ser139-phosphorylated histone H2A.X (pH2Ax), a hallmark of oxidative stress [30].

These experiments display that H_2_O_2_-treated C2C12 cells exhibit an increase in pH2Ax protein content, which was efficiently reversed by 3,4-DHPEA-EA(P), suggesting that DNA damage was caused by a ROS imbalance (Figure 3C). ROS production can induce a modulation of the redox state with a consequent modification of proteins, including carbonylation [31,32]. Thus, we measured the content of the protein oxidation in the total protein lysates.

As shown in Figure 3D, an increase in carbonylated proteins was observed in H_2_O_2_-treaded cells, which decreased significantly following a pre-treatment with 10 μM 3,4-DHPEA-EA(P), and thus confirming that an imbalance of the oxidative state is partially recovered by a pretreatment with an oleuropein compound. Subsequently, we analyzed the possible effects of 3,4-DHPEA-EA(P) on the activity of some of the ROS scavenging enzymes, such as Superoxide Dismutase 1 (SOD1), Catalase and Glutathione peroxidase 1 (Gpx1). As shown in Figure 3E, the natural compound is able to increase the levels of SOD1, Catalase and Gpx1 both in basal conditions and following treatment with H_2_O_2_. All these data suggest that 3,4-DHPEA-EA(P) exerts an antioxidant function under our experimental conditions.

### 2.4. 3,4-DHPEA-EA(P) Inhibits H_2_O_2_-Mediated Decrease of C2C12 Myocytes Viability by Inhibiting p-JNK Signaling Pathway

In understanding the mechanism(s) through which 3,4-DHPEA-EA(P) counteracts the H_2_O_2_-induced growth arrest and cell death of myocytes, we focused on the stress-activated c-Jun-N-terminal protein kinase (JNK) signaling pathway. In fact, it is known in the literature that the JNK/MAPK signaling pathway is significantly downregulated during myogenesis, by negatively adjusting the differentiation of skeletal muscle cells [33]. Thus, we initially evaluated whether 100 μM H_2_O_2_ was efficient in modulating the phospho-active form of JNK in our experimental system. We analyzed the protein content of p-JNK by Western blot analysis, using a phospho-specific antibody. As shown in Figure 4A, p-JNK was significantly increased after a 24 h treatment with 100 μM H_2_O_2_, and 10 μM 3,4-DHPEA-EA(P) efficiently counteracts this activation. We also measured p-JNK levels at short treatment times (1 and 3 h), but without observing any changes in its protein content (Figure 4B). We then attempted to delineate the JNK-pathway activated by the H_2_O_2_ treatment. Some studies explain that c-Jun is a transcription factor that regulates cell growth and the survival downstream JNK pathway [34]. As reported in Figure 4A, p-c-Jun accumulated after the H_2_O_2_ treatment in C2C12 myocytes and also in this case the 3,4-DHPEA-EA(P) was able to significantly reduce the phosphorylation of p-c-Jun, suggesting that the C2C12 growth arrest and cell dead proceeds via the JNK/c-Jun pathway. To confirm the role played by the p-JNK-p-c-Jun pathway in the H_2_O_2_-induced C2C12 growth arrest, we determined the effect of SP600125, a cell-permeable JNK inhibitor I and II. Figure 4C displays that SP600125 caused a significant reduction in JNK and c-Jun phosphorylation in H_2_O_2_-treated cells, confirming the involvement of the p-JNK signaling pathway in the H_2_O_2_-induced growth arrest.

### 2.5. 3,4-DHPEA-EA(P) Prevents C2C12 Atrophy Mediated by H_2_O_2_ Treatment

To demonstrate whether 3,4-DHPEA-EA(P) was able to function as an antioxidant also in relation to other oxidative stress inducers, we analyzed its effects in in vitro aged myotubes (day 8). In particular, C2C12 murine myoblasts were differentiated for 8 days and subsequently pre-treated with 10 μM 3,4-DHPEA-EA(P) and 100 μM H_2_O_2_. We have previously shown that in these conditions, myotubes express high levels of oxidative and nitrosative stress, as well as pro-inflammatory cytokines [35]. Similar to our above results, treatment with 3,4-DHPEA-EA(P) efficiently inhibited p-JNK, p-NF-kB and TNF-α compared to untreated day 8 myotubes (Figure 5A).

Finally, we checked the effect of 3,4-DHPEA-EA(P) on the myogenesis of C2C12 cells. Figure 5B shows that MyoD mRNA levels were significantly restored after the 3,4-DHPEA-EA(P) treatment, compared to the H_2_O_2_ treated myocytes, indicating a complete restoration of the differentiation process. This hypothesis was further confirmed by the analyses of Atrogin-1 (F-box protein 32) and Murf-1 (tripartite motif-containing 63), molecular factors involved in the atrophy process as muscle-specific E3 ubiquitin ligase that are increased during atrophy [36]. Figure 5C highlights a significant increase of their expression in H_2_O_2_-treated myocytes, suggesting that oxidative stress bring by H_2_O_2_ not only blocks the differentiation process but also triggers a degenerative process. The pre-treatment with 3,4-DHPEA-EA(P) prevents the induction of an atrophy-specific ubiquitin ligase, suggesting that this molecule could modulate the integrity of skeletal muscle.

## 3. Discussion

Metabolic alterations of skeletal muscle are associated with pathologies such as sarcopenia, muscular dystrophies and atrophy. These conditions are characterized by an accumulation of oxidative damage that may contribute to the loss of muscular tissue homeostasis. In healthy skeletal muscle, ROS are fundamental mediators of signaling pathways that have an impact on the proliferation, differentiation and apoptosis [37]. At low concentrations, ROS stimulates the healing and maintenance of muscle [38], but if ROS levels are excessively high, it can delay tissue repair and even worsen the injury, leading to atrophy [39]. To date, prevention and treatment for the reduction of oxidative stress under muscular atrophy or during the increase of the reactive species is not available. Nevertheless, antioxidant dietary supplementation was taken into consideration for the treatment of oxidative stress in muscles, as it is able to raise the levels of endogenous antioxidants or induce muscle repair [39,40].

In this context, it has been demonstrated that oleuropein, the main polyphenol of olive oil, and its derivatives had antioxidant and anti-inflammatory proprieties in skeletal muscle. In particular, the treatment with hydroxytyrosol (HT) increases creatine kinase activity and myosin heavy chain expression, which are indicators of muscle cell differentiation and contraction strength, respectively [41,42]. Moreover, oleuropein derivatives reduced the tumor necrosis factor-α (TNF-α)-induced downregulation of mitochondrial biogenesis, increasing the peroxisome proliferator-activated receptor-gamma coactivator (PGC-1α), mitochondrial complexes (I and II) and myogenin expression [43]. These data indicated that oleuropein improves the mitochondrial development and function in muscle cells under inflammatory stress.

In this study, we focused on the signaling pathways activated by the H_2_O_2_ treatment (used as model of oxidative stress) in C2C12 myocytes cell viability and differentiation. Our data demonstrated that high doses (100 μM) of H_2_O_2_ induce a significant inhibition of cell growth that resulted in the myocytes’ death through the activation of the canonical p-JNK/p-c-Jun signaling pathway. These data are supported by scientific literature demonstrating the implication of JNK’s pathway in H_2_O_2_-induced cell death [44]. H_2_O_2_ toxicity was dose-dependent, as already reported in the literature [45,46,47]. The failure to activate JNK at short treatment times could be due to the concentration and treatment time (perhaps too short) to which the differentiated myoblasts are exposed on day 2. Moreover, the activation of p-JNK/p-c-Jun signaling pathway was confirmed by experiments carried out with SP600125, which confirms the implication of JNK in our experimental system. It is known in the literature that this inhibitor seems to be selective also for the kinase upstream of JNK MKK4/7. For this reason, a decrease in JNK protein levels is observed following treatment with SP600125 [48].

Reactive oxygen species (ROS) production represents the initial step in the scale of events following the H_2_O_2_ treatment. The oxidative stress resulted in DNA damage, as demonstrated by the phosphorylation of histone pH2Ax, which controls the recruitment of the DNA repair machinery in response to DNA strand break, during the replication [49]. The involvement of ROS-mediated damage during the H_2_O_2_ cytotoxic action on myocytes was confirmed by using a treatment with the antioxidant catalase, which was able to efficiently abolish the myocytes’ growth arrest. Furthermore, we showed that the oxidative stress is able to block the differentiation process of myocytes, as shown by the drop of the MyoD expression. MyoD is a transcription factor implicated in the early stage of myogenic differentiation, and necessary for the progression of quiescent muscle satellite cells in the cell cycle [50]. In support of our data, recent studies have highlighted a correlation between nutrition and muscle homeostasis regulated by MicroRNAs (miRNAs). In particular, a poor nutritional state deficient in antioxidants, amino acids or iron determines a downregulation of miRNA133a/b by MyoD, leading to an increase in oxidative stress, inflammation and sarcopenia [51].

Moreover, tyrosol, a phenolic antioxidant present in olive oil, was demonstrated to be effective in inhibiting the H_2_O_2_-induced death of L6 muscle cells by regulating extracellular signal-regulated kinases (ERK), JNK and p38 MAPK [52]. Our findings reveal that 3,4-DHPEA-EA(P), a bioactive compound present in olive leaves, already suppressed H_2_O_2_-mediated cytotoxicity in C2C12 myocytes at a concentration of 10 μM, with a significant increase in viability. Moreover, our results showed that the 3,4-DHPEA-EA(P) treatment promotes the myogenesis process by restoring the expression of MyoD in H_2_O_2_-treated C2C12 myocytes.

In recent years, the role of MyoD in regulating the function of MiRNAs during muscle differentiation and sarcopenia has emerged. MiRNAs are small endogenous non-coding RNA molecules that play a central role in the development, differentiation and degeneration of skeletal muscle. For example, miRNA-233 is upregulated through MyoD, causing an inhibition of myoblasts proliferation, thus facilitating their differentiation [53]. Next to that, many authors have highlighted the existence of a correlation between myogenesis/sarcopenia, nutrition and regulation of MiRNAs mediated by MyoD. In particular, the authors highlight how a nutritional state poor in antioxidants, amino acids and glucose induces an activation of MyoD with consequent downregulation of miRNA133b and miRNA-206, which play a key role in the processes of development, differentiation and muscle regeneration. The miRNA133b and miRNA-206 downregulation is also connected to an increase in oxidative stress, inflammation and sarcopenia [54]. These data, together with our evidence, suggest that the correct use of natural molecules such as 3,4-DHPEA-EA(P) could regulate the levels of inflammation, oxidative stress that are observed during the degenerative processes and aging of the skeletal muscle.

In our experimental system, treatment with H_2_O_2_ induces an oxidative stress, which translates into an increase in carbonylated proteins. The pre-treatment with 3,4-DHPEA-EA(P) leads to a decrease in protein alterations and an increase in the main antioxidant enzymes (SOD1, Catalase and Gpx1). We assume that the mechanism involved goes through the activation of the transcription factor Nrf2, which we have previously shown to be implicated in the transcription of these enzymes [55]. It is known in the literature that Nrf2 is a transcription factor implicated in the protection against oxidative stress. In fact, many studies have shown that Nrf2 under different stimuli (i.e., treatment with antioxidants of natural origin) migrates inside the nucleus, recognizes the conserved consensus sequences on the target gene promoters, activating them [56]. Among these we find Catalase, SOD1 and the Gpx1 [57,58]. Similarly, our compound could mediate the detachment of Nrf2 from its inhibitor Keap1 by translocating it to the nucleus and mediating the activation of antioxidant genes downstream. At the moment our laboratory is in the testing phase of this pathway.

Among redox-sensitive factors able to modulate myocytes cycle progression and differentiation, MAPK are known to be activated in response to ROS production. In particular, it is known that p-JNK-p-c-Jun MAPK are negative regulators of myogenesis, playing a role opposite to that of p38 during differentiation [31], which instead is considered a pro-myogenic factor [53]. In accordance, we found that H_2_O_2_-induced oxidative stress activates the redox-sensitive p-JNK-c-Jun pathway exerting a pro-apoptotic role in muscle cells. Antioxidant treatment with 3,4-DHPEA-EA(P) not only restored redox balance, but also inhibits the p-JNK-c-Jun activation and phosphorylation, improving cell viability and differentiation of C2C12 myocytes.

As previously mentioned, an accumulation of oxidative damage could contribute to the loss of muscular homeostasis, function and atrophy. Interestingly, we showed that the increase of oxidative stress is associated with the increased marker of myotube atrophy, implying the occurrence of a degenerative process upon oxidative stress conditions. In contrast, the 3,4-DHPEA-EA(P) treatment prevents the muscle degeneration process, suggesting the potential use of this molecule in high oxidative stress conditions (i.e., physical exercise, sarcopenia and aging) or during atrophy or to regulate the expression of transcription factors such as MyoD and MiRNA dependent on it. These studies are being designed in our laboratory and the results could partially explain the importance of nutrition in modulating skeletal muscle epigenetics.

To conclude, our study highlights the antioxidant role of 3,4-DHPEA-EA(P) in C2C12 cells, as it is able to reduce intracellular levels of ROS, resulting in an inhibition of cell death and atrophy, conditions observed under increased oxidative stress. Future studies should be carried out to better clarify the effects of 3,4-DHPEA-EA(P) on a cellular level and in in vivo models, in order to develop new therapeutic strategies for the treatment of muscular diseases related to the increase of oxidative stress.

## 4. Materials and Methods

### 4.1. Cell Cultures and Treatments

The murine skeletal muscle C2C12 cells, obtained from the European Collection of Cell Cultures (Salisbury, UK), were cultured in growth mediums composed of Dulbecco’s Modified Eagle’s Medium (DMEM) supplemented with a fetal bovine serum (FBS, 10%), penicillin/streptomycin (100 U/mL) and glutamine (2 mM) (Lonza Sales, Basel, Switzerland), and maintained at 37 °C in an atmosphere cointaining 5% CO_2_ in the air. C2C12 myocytes were plated at 80% of the confluence and cultured in a growth medium for 24 h. To induce differentiation, cells were washed with PBS and the growth medium was replaced with a differentiation medium (DM), which contained 2% of heat inactivated horse serum (Lonza, ECS0090D, Basel, Switzerland) for 2 days [59].

Treatments with H_2_O_2_ were performed with different concentrations ranging from 20 to 100 μM after the change with DM (at day 2 of the differentiation). The concentration of 100 μM H_2_O_2_ was designated for all experiments as it provided the most significant degree of cell growth arrest. 3,4-DHPEA-EA and the peracetylated derivative, 3,4-DHPEA-EA(P) were used at a concentration of 10 μM (1 h before H_2_O_2_ treatment), and maintained throughout the experiment. Treatments with 3,4-DHPEA-EA and 3,4-DHPEA-EA(P) were also performed with different concentrations ranging from 5 to 100 μM. The concentrations of 10 μM 3,4-DHPEA-EA and 3,4-DHPEA-EA(P) were chosen because they provided the most significant degree of cell viability recovery. As a control, equal amounts of DMSO (0.1%) were added to untreated cells. In the indicated experiments, catalase was added 1 h prior to the H_2_O_2_ treatment at a concentration of 1 μM, and maintained throughout the experiment. In the indicated experiments, treatments with cell-permeable JNK inhibitor I and II (SP600125) (Invivogen) were performed at a concentration of 10 μM because lower concentrations did not show significant inhibition, and higher concentrations proved to be toxic [60]. SP600125 was added concomitantly with H_2_O_2_ and maintained throughout the experiments.

### 4.2. Analysis of Cell Viability and Proliferation

After the trypsinization, adherent and detached cells were combined, washed with PBS and directly counted with an optical microscope on a hemocytometer after the Trypan Blue staining. Alternatively, cell proliferation was measured by using a MTS “Cell Titer 96^®^ Aqueous One Solution Cell Proliferation assay” kit (Promega, Madison-Winsconsin), following to the manufacturer’s instructions. Cell proliferation was also measured with a “Cell Proliferation kit” (Buckinghamshire, UK), based on the immunocytochemical detection of 5-bromo-2′-deoxyuridine (BrdU) incorporated in the cellular DNA of proliferating cells. Cells were stained as previously described [61].

### 4.3. RT-qPCR Analysis

A TRI Reagent (Sigma-Aldrich) was used to extract Total RNA, useful for the retro-transcription. qPCR was performed in triplicate by using validated qPCR primers (BLAST), Ex TAq qPCR Premix (Lonza Sales) and the Roche Real Time PCR LightCycler II (Roche Applied Science, Monza, Italy). mRNA levels were normalized to RPL, and the relative mRNA levels were determined by using the 2^−ΔΔCt^ method [45]. The primer sequences are listed in Table 1.

### 4.4. Preparation of Cell Lysates and Western Blot Analyses

Cell pellets were suspended in the RIPA buffer (50 mM Tris-HCl, pH 8.0, 150 mM NaCl, 12 mM deoxycholic acid, 0.5% Nonidet P-40 and protease inhibitors). Protein samples were resolved by SDS-PAGE and subjected to Western blotting, as previously described [36]. Nitrocellulose membranes were stained with primary antibodies against Tubulin (1:1000), p-H2A.x (1:1000), p-JNK (1:1000), JNK (1:1000), p-c-Jun (1:1000), TNFα (1:1000) and p-NF-kB (1:1000). Next, the membranes were incubated with the apposite horseradish peroxidase conjugated secondary antibody, and immunoreactive bands were detected by using a Fluorchem Imaging System upon staining them with an ECL Select Western Blotting Detection Reagent (GE Healthcare, Pittsburgh, PA, USA; RPN2235). The immunoblots reported in the figures are representative of at least four independent experiments, which led to similar results. Tubulin was used as loading control.

Proteins were assayed with the Lowry method [62].

### 4.5. Determination of Protein Carbonylation

Carbonylated proteins were detected by using the OxyBlot Kit (Millipore, S7150, Burlington, MA, USA) as previously described [22]. Briefly, 20 μg of total proteins were reacted with 2,4 dinitrophenylhydrazine (DNP) for 15 min at 25 °C. Samples were resolved on 10% SDS-polyacrylamide gels and DNP-derivatized proteins were identified by Western blot analysis, using an anti-DNP antibody and an appropriate horseradish peroxidase-conjugated secondary antibody.

### 4.6. Determination of ROS

ROS were detected by cytofluorimetric analysis following incubation for 1 h before the end of the experiments, at 37 °C with 50 μM DCF-DA. After treatment, the cells were scraped, washed and resuspended in a PBS. The fluorescence intensities of 10,000 cells from each sample were performed by using a FACScalibur instrument (Beckton and Dickinson, San Josè, CA, USA) and analyzed using the WinMDI 2.8 software. Otherwise, the cells were incubated with DHE for 30 min at 37 °C to measure O_2_^−^ production. Subsequently, the cells were collected and used for cytofluorimetric analyses through a FACScalibur instrument (BD Bioscience, Franklin Lakes, NJ, USA).

### 4.7. Statistical Analysis

Data were expressed as means ± standard deviation (S.D.). Statistical evaluation was conducted by ANOVA, followed by the post hoc Student-Newman-Keuls. Differences were considered to be significant at *p* < 0.05.

### 4.8. Microwave-Assisted Extraction of Oleuropein from Olive Leaves

One hundred grams of dried and milled olive leaves and 800 mL of water were placed in a Pyrex round-bottom flask equipped with a jacketed coiled condenser in a domestic microwave oven. The sample irradiated for 10 min at 800 W. Leaves were filtered and the organic solution was dried under pressure. The mixture was washed and filtered with acetone to eliminate the solid residue, and the solution was evaporated under pressure. The crude product was purified by liquid chromatography on a Supelco versa-flash station (Supelco Inc., Bellefonte, PA, USA) equipped with a silica cartridge and a mixture of CH_2_Cl_2_/MeOH 8:2 as mobile phase to obtain pure oleuropein. The purity was determined by LC-QTOF-MS and ^1^H-NMR.

LC-MS (*m*/*z*) calcd for C_25_H_32_O_13_ [M]^+^ 540.1843, measured 540.1828, found 563.1726 C_25_H_32_NaO_13_ [M + Na]^+^.

^1^H-NMR: Spectral data were in accordance with the literature [63].

### 4.9. Synthesis of 3,4-DHPEA-EA and 3,4-DHPEA-EA (P)

3,4-DHPEA-EA (methyl-4-(2-(3,4-dihydroxyphenethoxy)-2-oxoethyl)-3-formyl-2-methyl-3,4-dihydro-2H-pyran-5-carboxylate) was obtained by catalytic hydrolysis of oleuropein previously extracted from dried olive leaves. It was obtained as yellow oil (70% yield). DHPEA-EA(P) (methyl-4-(2-(3,4-dihydroxyphenethoxy)-2-oxoethyl)-3-formyl-2-methyl-3,4-dihydro-2H-pyran-5-carboxylate– peracetylated) was obtained by catalytic acetylation [7]. It was obtained as a yellow solid (79% yield). ^1^H and ^13^C NMR spectra of 3,4-DHPEA-EA and 3,4-DHPEA-EA (P) were in agreement with the literature data [7]. ^1^H and ^13^C NMR spectra were recorded at 300 and 75 MHz respectively in CDCl_3_, using tetramethylsilane (TMS) as internal standard on a Bruker ACP 300 MHz instrument (Bruker, Billerica, MA, USA).

The olive leaves, samples used for the extraction of oleuropein in order to obtain 3,4-DHPEA-EA and 3,4-DHPEA-EA (P), came from the *Coratina* cultivar of the Olea *europaea*, collected from plants belonging to the olive grove of the CREA, Research Centre for Olive, Citrus and TreeFruit, Rende, Cosenza, Italy, placed at 204 m a.s.l. (39 22′17.681″ N, 16 13′58.342″ E). The sample was dried for 48 h at 50 °C, powdered and preserved at room temperature until use.

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
