# Peer review of "Oleuropein Aglycone Peracetylated (3,4-DHPEA-EA(P)) Attenuates H2O2-Mediated Cytotoxicity in C2C12 Myocytes via Inactivation of p-JNK/p-c-Jun Signaling Pathway"

_molecules, 2020, doi:10.3390/molecules25225472_

Round 1

Reviewer 1 Report

Authors investigated olive leave extract chemical, oleuropein can reduce h2o2 induced cytotoxicity in myocytes. They investigated molecular pathways of this protection and identified oleuropein inactivate JNK pathway. 

Generally, I have no big concerns for this manuscript. But I have several minor comments.

Scheme 1 words should be enlarged

First 2 paragraphs line171-182 may not be the best suited for results section.

Generally it is difficult to read small fonts in figures. Please enlarge them.

Figure 5, I don't know the meaning of circle. It was not commented in the legend. line 356 H2O2's 2 should be lower case.

I don't see the link to the supplementary materials. I recommend authors should transfer to main manuscript rather than keeping them in supplementary materials.

Author Response

Dear Reviewer,

Thank you for your email, referring to the reviewer’s process of our manuscript entitled “Oleuropein aglycone peracetylated (3,4-DHPEA-EA(P)) attenuates H2O2-mediated cytotoxicity in C2C12 myocytes via inactivation of p-JNK/p-c-Jun signaling pathway”. We really appreciated the comments that give us the possibility to ameliorate the value of the manuscript.

Sincerely yours

Monica Nardi and Sara Baldelli

Next, we explained point-by-point the details of the revisions in the manuscript and responses to the reviewers' comments.

  1. Scheme 1 words should be enlarged

Words in the Scheme 1 enlarged

  1. First 2 paragraphs line171-182 may not be the best suited for results section.

First 2 paragraphs line171-182 transferred in the introduction section

  1. Generally it is difficult to read small fonts in figures. Please enlarge them.

We have now enlarged the characters of all the figures.

  1. Figure 5, I don't know the meaning of circle. It was not commented in the legend. line 356 H2O2's 2 should be lower case.

We have now added the meaning of the circle and changed it to lowercase H2O2.

  1. I don't see the link to the supplementary materials. I recommend authors should transfer to main manuscript rather than keeping them in supplementary materials.

The supplementary materials transferred to main manuscript. We addes the new paragraph 2.8 Microwave-Assisted Extraction of Oleuropein from Olive Leaves in the main manuscript.

Thank you for the patience you have had in waiting for my answer.

Reviewer 2 Report

The paper shows a role of the olive leaf extract 3,4-DHPEA-EA(P) in protection against hydrogen peroxide.  Aspects of the paper could be improved to make a stronger case and highlight the novel actions of the compound.

Specific comments for revision:

Major

  1. Abstract: the last sentence of the abstract suggests that 3,4-DHPEA-EA(P) is an efficient antioxidant and that it also exerts effects via epigenetic mechanisms. Neither of these is addressed by data in the current student, so it’s inappropriate to mention them in the abstract.

  1. Please consider moving lines 79-84 to the Discussion. They’re inappropriate in the paragraph summarizing findings

  1. Line 84: there’s no evidence provided in the paper that the compound acts as a powerful antioxidant. You could say something like it protects against hydrogen peroxide challenge, but your study does not show that it does so by acting as an antioxidant.

  1. Line 303 and fig 4: the JNK inhibitor appears to also inhibit the upstream kinase for JNK (i.e. it suppresses JNK phosphorylation). This should be addressed in the Discussion by presenting literature on any off-target effects of the inhibitor and how this might affect the interpretation of the current study.  A better means of demonstrating a role in JNK in response to hydrogen peroxide (e.g. in terms of cell death) should be provided (either with new data or by citing the literature).

  1. Line 345 again there’s a statement that the compound acts as an antioxidant. Evidence should be provided from the literature or from new data on in vitro (cell-free) antioxidant action of the compound, or the statements about its antioxidant action should be omitted.

  1. The data in figure 3A are potentially the most important in the paper. The compound increases expression of antioxidant genes. Please expand the discussion of this to include the transcriptional processes known to control expression of these genes and how the compound might be increasing their expression.  This antioxidant-independent mechanism of action could be an important quality of the compound.

Minor

  1. Line 41: it appears like there’s a word missing in “oleuropein is a potent superoxide anion.”

  1. Line 47: please spell out EVOO

  1. Line 63: “decreasing insulin levels at muscular level” is incorrect. Please check the reference (#11).  Possibly you mean “decreasing insulin resistance in skeletal muscle.”

  1. Lines 77, 386: do you mean “oxidative stress” instead of “oxidative burst?”

  1. Lines 108-109: this sentence is unfinished

  1. Please discuss the lack of short-term changes in P-JNK when exposed to H2O2 for 1 or 3 h. Was this expected?

  1. Use of catalase in the medium is not a strong experimental design. It essentially would almost instantly catalyze removal of H2O2, making this treatment the same as if H2O2 weren’t in the medium.

Author Response

Dear Reviewer,

Thank you for your email, referring to the reviewer’s process of our manuscript entitled “Oleuropein aglycone peracetylated (3,4-DHPEA-EA(P)) attenuates H2O2-mediated cytotoxicity in C2C12 myocytes via inactivation of p-JNK/p-c-Jun signaling pathway”. We really appreciated the comments that give us the possibility to ameliorate the value of the manuscript.

Sincerely yours

Monica Nardi and Sara Baldelli

Next, we explained point-by-point the details of the revisions in the manuscript and responses to the reviewers' comments.

Major 

  1. Abstract: the last sentence of the abstract suggests that 3,4-DHPEA-EA(P) is an efficient antioxidant and that it also exerts effects via epigenetic mechanisms. Neither of these is addressed by data in the current student, so it’s inappropriate to mention them in the abstract.

We have now removed this sentence from the abstract. 

  1. Please consider moving lines 79-84 to the Discussion. They’re inappropriate in the paragraph summarizing findings.

We have now moved the sentence into the discussion section.  

  1. Line 84: there’s no evidence provided in the paper that the compound acts as a powerful antioxidant. You could say something like it protects against hydrogen peroxide challenge, but your study does not show that it does so by acting as an antioxidant.

We have now changed the sentence.

  1. Line 303 and fig 4: the JNK inhibitor appears to also inhibit the upstream kinase for JNK (i.e. it suppresses JNK phosphorylation). This should be addressed in the Discussion by presenting literature on any off-target effects of the inhibitor and how this might affect the interpretation of the current study.  A better means of demonstrating a role in JNK in response to hydrogen peroxide (e.g. in terms of cell death) should be provided (either with new data or by citing the literature).

We have now specified, based on the literature, why the kinase upstream of JNK is partially inhibited. this explanation was included in the discussion. [51]. Moreover, we also mentioned the role of JNK in response to H2O2 [54]. In the discussion we reported “These data are supported by scientific literature demonstrating the implication of JNK's pathway in H2O2-induced cell death [51]. H2O2 toxicity was dose-dependent as already reported in the literature [32, 52, 53]. The failure to activate JNK at short treatment times could be due to the concentration and treatment time (perhaps too short) to which the differentiated myoblasts are exposed on day 2. Moreover, the activation of p-JNK/p-c-Jun signaling pathway was confirmed by experiments carried out with SP600125, which confirms the implication of JNK in our experimental system. It is known in the literature that this inhibitor seems to be selective also for the kinase upstream of JNK MKK4/7. For this reason a decrease in JNK protein levels is observed following treatment with SP600125 [54]”.

  1. Line 345 again there’s a statement that the compound acts as an antioxidant. Evidence should be provided from the literature or from new data on in vitro (cell-free) antioxidant action of the compound, or the statements about its antioxidant action should be omitted.

We have now eliminated this statement. 

  1. The data in figure 3A are potentially the most important in the paper. The compound increases expression of antioxidant genes. Please expand the discussion of this to include the transcriptional processes known to control expression of these genes and how the compound might be increasing their expression.  This antioxidant-independent mechanism of action could be an important quality of the compound.

 We have now better explained the mechanism of gene control of antioxidant enzymes. We reported “It is known in the literature that Nrf2 is a transcription factor implicated in the protection against oxidative stress. In fact, many studies have shown that Nrf2 under different stimuli (i.e. treatment with antioxidants of natural origin) migrates inside the nucleus, recognizes the conserved consensus sequences on the target gene promoters, activating them [62]. Among these we find Catalase, SOD1 and the Gpx1 [63,64]. Similarly, our compound could mediate the detachment of Nrf2 from its inhibitor Keap1 by translocating it to the nucleus and mediating the activation of antioxidant genes downstream.

Minor 

  1. Line 41: it appears like there’s a word missing in “oleuropein is a potent superoxide anion.” 

We changed the sentence with “oleuropein is a potent scavenger of superoxide anion radicals and of other reactive oxygen species (ROS)”

  1. Line 47: please spell out EVOO

We reported “extra-virgin olive oil (EVOO)”

  1. Line 63: “decreasing insulin levels at muscular level” is incorrect. Please check the reference (#11).  Possibly you mean “decreasing insulin resistance in skeletal muscle.”

We changed the sentence with

  1. Lines 77, 386: do you mean “oxidative stress” instead of “oxidative burst?”

We have now replaced burst with stress.

  1. Lines 108-109: this sentence is unfinished 

We finisched the sentence

  1. Please discuss the lack of short-term changes in P-JNK when exposed to H2O2 for 1 or 3 h. Was this expected?

The failure to activate JNK at short treatment times could be due to the concentration and treatment time (perhaps too short) to which the differentiated myoblasts are exposed on day 2. In fact, once differentiated, these cells seem much less susceptible to insults than undifferentiated myoblasts. 

  1. Use of catalase in the medium is not a strong experimental design. It essentially would almost instantly catalyze removal of H2O2, making this treatment the same as if H2O2 weren’t in the medium.

Usually catalase is used as a positive control to demonstrate the involvement of H2O2 in mediating a given cellular response. We believe that this treatment can help in understanding and verifying the experiments.

REFERENCES

  1. Zhang, S.; Lin, Y.; Kim, Y.S.; Hande, M.P.; Liu, Z.G.; Shen, H.M. c-Jun N-terminal kinase mediates hydrogen peroxide-induced cell death via sustained poly(ADP-ribose) polymerase-1 activation. Cell death and differentiation 2007, 14, 1001-1010
  2. Kim, H.; Lee, K. II.; Jang, M.; Namkoong, S.; Park, R.; Ju, H.; Choi, I.; Oh, W. K.; Park, J. Conessine Interferes with Oxidative Stress-Induced C2C12 Myoblast Cell Death through Inhibition of Autophagic Flux. PLoS One, 2016, 11(6), 1-9.
  3. Santa-Gonzalez, G.A.; Gomez-Molina, A.; Arcos-Burgos M.; Meyer, J.N.; Camargo, M. Distinctive adaptive response to repeated exposure to hydrogen peroxide associated with upregulation of DNA repair genes and cell cycle arrest. Redox Biol, 2016, 9, 124-133
  4. Bennett, B.L.; Sasaki, D.T.; Murray, B.W.; O'Leary, E.C.; Sakata, S.T.; Xu, W.M.; Leisten, J.C.; Motiwala, A.; Pierce, S.; Satoh, Y., et al. SP600125, an anthrapyrazolone inhibitor of Jun N-terminal kinase. P Natl Acad Sci USA 2001, 98, 13681-13686
  5. Tonelli, C.; Chio, I.I.C.; Tuveson, D.A. Transcriptional Regulation by Nrf2. Antioxid Redox Sign 2018, 29, 1727-1745.
  6. Dong, J.; Sulik, K.K.; Chen, S.Y. Nrf2-mediated transcriptional induction of antioxidant response in mouse embryos exposed to ethanol in vivo: Implications for the prevention of fetal alcohol spectrum disorders. Antioxid Redox Sign 2008, 10, 2023-2033.
  7. Zou, Y.; Wang, J.; Peng, J.; Wei, H.K. Oregano Essential Oil Induces SOD1 and GSH Expression through Nrf2 Activation and Alleviates Hydrogen Peroxide-Induced Oxidative Damage in IPEC-J2 Cells. Oxid Med Cell Longev 2016, 1-13.

Round 2

Reviewer 2 Report

.